# Melanoma in Peru: 1000 patients and 10 years of experience

Gonzalo Ziegler-Rodriguez[1,2,3], Gabriel De La Cruz-Ku[3]*, Anshumi Desai[4], Luis Piedra Delgado[3], Silvana Maldonado[3], Jiddu Antonio Guart[5], Camila Franco[6], Sheyla Diaz-Mora[1], Sheila Vilchez Santillan[1], Jorge Dunstan Yataco[1], Juan Haro-Varas[1], Jose Galarreta Zegarra[1], Sandro Casavilca Zambrano[1], Jose Cotrina Concha[1]

1 Instituto Nacional de Enfermedades Neoplásicas, Lima, Perú, 2 Melanoma and Skin Cancer Unit, Clinica Ziegler, Lima, Peru, 3 Universidad Científica del Sur, Lima, Perú, 4 Department of Plastic Surgery, University of Miami Miller School of Medicine, Miami, Florida, United States of America, 5 University of Massachusetts Medical School, Worcester, Massachusetts, United States of America, 6 Department of Surgery, University of Texas Medical Branch at Galveston, Galveston, Texas, United States of America

* gabrieldelacruzku@gmail.com

## Abstract

### Background

Epidemiological studies identify characteristics of melanoma based on race and ethnic groups. Previous reports have revealed that the Hispanic/Latino population with melanoma has different characteristics compared to other races. We aimed to describe the sociodemographic, clinicopathological features, and long-term outcomes such as event-free survival (EFS) and overall survival (OS) Peruvian population with melanoma.

### Methods

We conducted a retrospective cohort study of new cases diagnosed and treated at a single tertiary institution from 2010 to 2019. Survival analysis included patients with stages I-IV. Multivariable Cox Regression analysis was used to assess prognostic factors.

### Results

A total of 1136 patients were included, the median age at diagnosis was 63 years old (6−97 years), the majority were males (51.5%), resided in a non-metropolitan area (77.7%), had a primary lesion in lower extremities (75.5%), ulceration (52.4%), and acral lentiginous was the most common histological subtype (38.1%). The mean Breslow was 8.88, and 6.62 number of mitoses per mm. Stage T3-4 was present in 70.4%, while Stage III was the most frequent AJCC Stage (36.5%). Regarding adjuvant therapy, 13.4% received chemotherapy, 3.6% radiotherapy, and 13.6% interferon. No patients received immunotherapy. At 5-year follow-up, the EFS was

**Data availability statement:** All relevant data are within the manuscript and its Supporting Information files.

**Funding:** The author(s) received no specific funding for this work.

**Competing interests:** The authors have declared that no competing interests exist.

20%, while the OS was 36%. Prognostic factors of worse EFS were greater age (HR = 1.020), male sex (HR = 1.369), greater mitotic rates (HR = 1.012), ulceration (HR = 1.564), positive lymph nodes (HR = 2.58), while primary lesion in upper extremities (HR = 0.518) was associated to better EFS. Prognostic factors of worse OS were greater age (HR = 1.024), greater mitotic rate (HR = 1.013), ulceration (HR = 1.72), and positive lymph nodes (HR = 3.231).

## Conclusions

The sociodemographic features of Peruvian patients with melanoma are different from other populations. There were higher stages without access to immunotherapy, therefore EFS and OS rates were worse than international reports, while prognostic factors were similar. Urgent healthcare policy changes are needed as is access to immunotherapy.

## Introduction

The incidence of melanoma continues to rise globally due to increased exposure to ultraviolet radiation from sunlight [1]. This neoplasm is one of the most aggressive forms of skin cancer with a poor prognosis when diagnosed at advanced stages [2]. While most melanoma cases occur in individuals with fair skin, recent studies have highlighted significant disparities in the clinical presentation and outcomes of melanoma among different racial and ethnic groups [3,4].

Several differences have been noted between different populations and regions, such as the trunk is the most common location for the primary lesion in white populations, whereas non-white populations tend to exhibit lesions more frequently on the lower extremities, including the foot, leg, and acral areas [5]. Additionally, superficial spreading melanoma (SSM) was the most common subtype for white and Hispanic populations, while acral lentiginous melanoma (ALM) was more prevalent in Asian and African American populations. At diagnosis, patients were more likely to present with local stage disease, whereas non-white populations had a higher percentage of regional or distant stages at presentation [5,6]. A systematic review of more than three million patients showed that there is a higher risk of mortality from any cause in black patients compared to non-Hispanic white patients. Similarly, Hispanic patients have shown to have worse survival outcomes [7].

In Latin America, there is scarce literature about sociodemographic features and outcomes, some of these found significant differences with worse outcomes [8–10]. However, as demonstrated in previous studies different regions can have different clinical presentation, tumor biology, and therefore outcomes that can stem from multiple factors [11,12]. Expanding research in the Hispanic Latin population is essential to address the disparities and underrepresentation of these patients in melanoma studies. Therefore, this study aimed to describe the sociodemographic and clinicopathological features, as well as the long-term outcomes, of a large-scale Peruvian population with melanoma.

## Materials and methods

### Study design and patients

We performed a retrospective cohort study. We reviewed medical records from patients diagnosed with melanoma and treated at the "Instituto Nacional de Enfermedades Neoplasicas" in Lima, Peru, from 2010 to 2019. The patients were followed up until October 2024. The medical records were accessed and data was collected from October 2022 to October 2024. Our inclusion criteria were: patients who were 18 years old and older, patients with a confirmed diagnosis of melanoma, stages I to IV at diagnosis, and all specimens confirmed by the Pathology department of the institute, with all cases reviewed by at least two pathologists. Tumor thickness was assessed according to Breslow depth. In cases where histopathological evaluation alone was insufficient, immunohistochemistry markers were employed to support the diagnosis of melanoma. Gene expression signatures are not available in public institution in our country. We excluded the patients who were lost to follow-up, mucosal melanoma, melanoma of the head and neck, and patients treated at another institution. Melanoma of the head and neck was not included due to these patients are evaluated and taken care of by another department different than Surgical Oncology and sentinel lymph node biopsy is not routinely performed which is important for staging purposes. Moreover, the management based on international guidelines is not completely reliable, therefore to avoid any bias in our study, we excluded them. The reason for not including more recent years was to ensure a minimum five-year follow-up period (2019–2024); therefore, the last year included was 2019.Immunotherapy is not available in public institutions in Peru such as the institution mentioned above. No patients involved in this study were part of national or international clinical trials.

### Variables

Sociodemographic, clinicopathological, and therapeutic characteristics were collected. Time to diagnosis refers to the time interval since the patient first noticed a skin lesion to the appointment when melanoma was diagnosed. Residence was classified according to the United States Department of Agriculture guidelines [13], metropolitan areas were defined as regions with central counties containing urban areas of 50,000 people or more, while non-metropolitan areas referred to less accessible regions without urban centers. OS was defined as the timeframe from the diagnosis of the primary tumor until death or end of the study. Event-free survival was defined for patients who underwent surgery since the date of surgery until the locoregional or distant relapse, death or end of the study; while those who did not undergo surgery at the time from diagnosis to progression of the disease, death, or end of the study.

Melanoma recurrence was identified with biopsy, ultrasound, Positron Emission Tomography-Computed Tomography, or Magnetic resonance imaging. All patients were classified according to the eighth edition of the American Joint Committee on Cancer Staging Manual (AJCC) [14], and treated according to the National Comprehensive Cancer Network (NCCN) guidelines at a comprehensive tertiary cancer center [15]. For patients who were diagnosed before the 8th edition of the AJCC was published, all cases were reviewed based on clinical and pathological data to perform an accurate staging re-classification. After surgical intervention, all patients were referred to medical oncology for further adjuvant therapy, while patients with metastatic disease were seen by medical oncology. The follow up of the patients was done according to NCCN guidelines. After completion of treatment, for stages I-IIA, patients were followed up every 6 months for five years and then annually as clinically indicated. For patients with stage IIB to IV, these were followed every three months for a total of two years, then every six months for three additional years, then annually depending on the clinical status of the patient. For patients with stage IIB to IV, every patient has scheduled computed tomography (CT) scans with intravenous contrast and ultrasounds of the adjacent lymph node regions every 12 months for three years, then additional CT scans or ultrasounds are performed if the patient presents symptoms.

## Statistical analysis

Descriptive statistics were used to assess sociodemographic, clinical, surgical, and therapeutic characteristics. We used mean and standard deviation (SD) or median with interquartile range (IQR) for quantitative variables; frequencies and percentages were used to assess qualitative variables. Kaplan-Meier method was used to calculate survival rates, while the Log-rank test was used to assess the differences between categories. Survival outcomes were evaluated at 3- and 5-years follow-ups. Patients who were lost to follow-up were excluded from the analysis. Missing data were reported in the tables and not included in our analyses. Missing values were reported as NULL in the dataset (S1-S2 Files). Prognostic factors were reported with Hazar ratios (HR) which were estimated with multivariable Cox regression analysis. To address any potential bias in our analyses, all population available was included in our analyses, no sample was calculated. A 95% confidence interval was used and a p-value <0.05 was considered statistically significant. The Statistical Package for Social Sciences (SPSS) software version 28.0 was used for all the analyses.

## Ethics

This study was approved by the Institutional Review Board (IRB) from the "Instituto Nacional de Enfermedades Neoplasicas" (INEN 22–36). Due to the retrospective design of our study, the consent was waived by the ethics committee. Data was collected confidentially by the principal investigator after obtaining the authorization to access the medical records by the IRB from the institution. The study was funded by all the authors, no financial support was received from any financial institution or external entity. All data was handled with strict confidentiality and only used for study purposes, it was de-identified and clinical information was codified. S1-S2 Files.

## Results

From a total of 1,734 patients diagnosed with melanoma, 1,136 met the eligibility criteria (Fig 1). The median age at diagnosis was 63 years (range: 51–72 years). 11.2% were younger than 40 years old (n = 127), while the majority of the population was between 60–69 years old (25.4%). The population was predominantly male with 51.5% (n = 585), only 15.5% (n = 172) living in Lima, the capital, and the majority, 77.7% (n = 883), living in other regions of Peru. Occupationally, 81.9% (n = 876) of patients worked indoors, and 18.1% (n = 193) were engaged in outdoor work (Table 1). Regarding the prevalence of melanoma over the years, the prevalence increased from 9.9% in 2010 to a peak of 12.9% in 2013, followed by a gradual decline in subsequent years, reaching 7.1% in 2019 (Fig 2).

The median time to diagnosis was 12 months (IQR: 6–24 months). 65% noted their lesions for more than 6 months before. The majority of the population (75.5%) had the primary lesion on the lower extremities. Clinical features of the lesions showed asymmetry in 68.8% (n = 782), irregular edges in 60.9% (n = 690), heterogenic color in 68.0% (n = 772), and ulceration in 52.4% (n = 595). There were 6.9% (n = 78) with satellite lesions, while lymph node assessments indicated 18.2% (n = 206) had regional palpable lymph nodes, and 19.6% (n = 222) had matted lymph nodes (Table 2).

Surgical management of the primary lesions involved wide local excision in 55.4% (n = 482), amputation of fingers or toes in 25.5% (n = 222), and extremity amputation in 3.0% (n = 26). Sentinel lymph node surgery was performed in 54.5% (n = 485), and a complete lymph node dissection was performed in 26.1% (n = 232). The mean number of lymph nodes removed was 5 (IQR: 2–13), and the mean number of positive lymph nodes was 1 (IQR: 0–2). Surgery for metastasis was performed in 0.6% (n = 7). The most common histological subtype was acral lentiginous melanoma in 38.1% (n = 422), followed by nodular melanoma in 13.9% (n = 154). Lymphovascular invasion was present in 12.1% (n = 107), and perineural invasion was present in 14.9% (n = 119). The mean Breslow thickness was 8.88 mm [SD 9.59 mm], and the mean mitotic rate was 6.62 mitoses/mm$^2$ [SD 7.42]. A total of 70.4% of patients were stage T3-4 and 63% were lymph node positive at diagnosis. Most of the patients were stage III (36.5%) at diagnosis, followed by stage II (23.9%) and IV

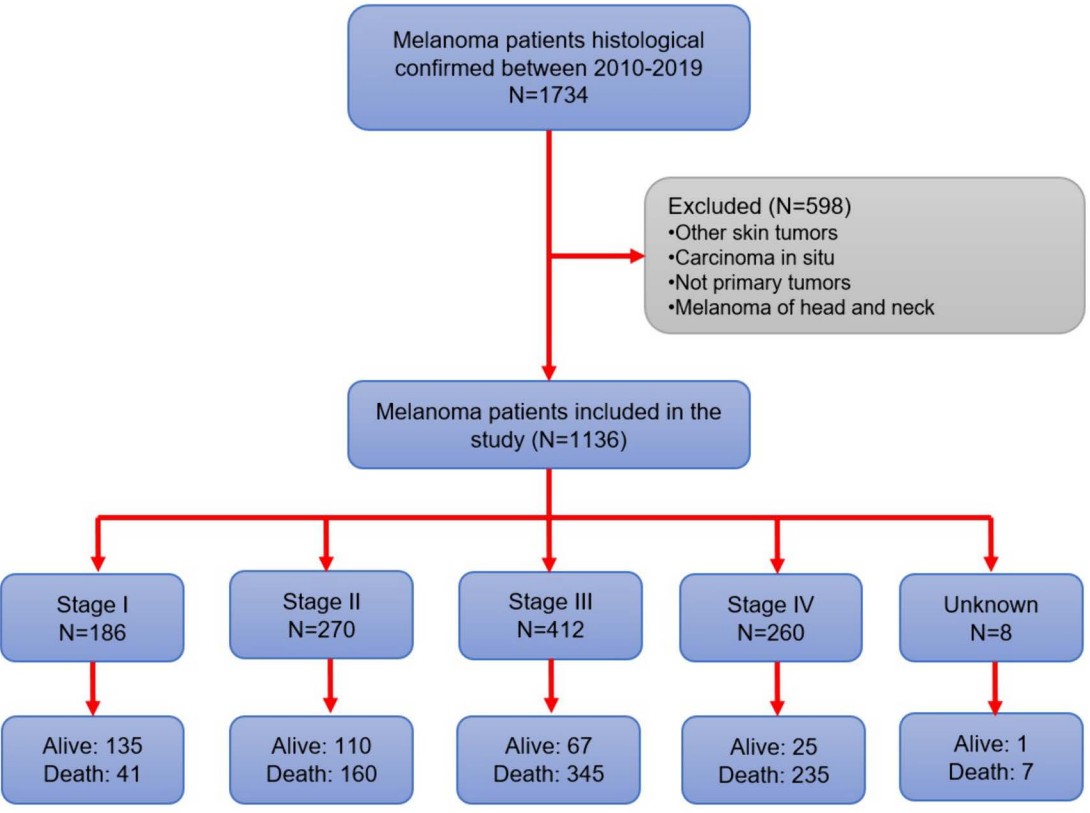

**Fig 1. Graphic schematization of patients inclusion in the study.**

(23.0%). Regarding adjuvant therapy, 13.4% (n = 152) received adjuvant chemotherapy, 3.6% (n = 41) underwent adjuvant radiotherapy, and 13.6% (n = 155) received adjuvant interferon therapy. The most common sites of distant recurrence or progression included lung (12.1%, n = 138), brain (7.2%, n = 82), liver (3.4%, n = 39), and bone (2.2%, n = 25) (Table 3).

Our study had a median follow-up of 122 months (10.2 years) and an attrition rate of 5.1% and 17.5% at 5 and 10 years, respectively. The EFS rates at 3- and 5- years follow-up were 77% and 67% for stage I, 26% and 51% for stage II, 12% and 23% for stage III, and 1% and 3% for stage IV, respectively (p < 0.001). While the OS rates at 3- and 5- years follow-up were 85% and 92% for stage I, 51% and 73% for stage II, 22% and 36% for stage III, and 6% and 9% for stage IV, respectively (p < 0.001) (Table 4, Figs 3 and 4).

Multivariable analysis showed that prognostic factors of worse EFS were greater age (HR = 1.020, 95% CI: 1.013–1.028, p < 0.001), male sex (HR = 1.369, 95% CI: 1.133–1.654, p = 0.001), greater mitotic rates (HR = 1.012, 95% CI: 1.001–1.024, p = 0.036), presence of ulceration (HR = 1.564, 95% CI: 1.217–2.009, p < 0.001), NOS and other subtypes compared to acral and nodular histology (HR = 1.357, 95% CI: 1.091–1.687, p = 0.006), positive lymph nodes (HR = 2.580, 95% CI: 2.091–3.182, p < 0.001); while factors associated with better EFS were primary lesion located in upper extremities compared to trunk (HR = 0.518, 95% CI: 0.335–0.801, p = 0.003). On the other hand, prognostic factors of worse OS were greater age (HR = 1.024, 95% CI: 1017 = 1.031, p < 0.001), greater mitosis per mm2 (HR = 1.013, 95%CI: 1.001–1.025, p = 0.032), the presence of ulceration (HR = 1.621, 95%CI: 1.251–2.099, p < 0.001), and positive lymph nodes (HR = 3.231, 95%CI: 2.597–4.020, p < 0.001) (Table 5).

**Table 1. Sociodemographic features of the Peruvian population diagnosed with melanoma.**

| Variables | Total population (N = 1136) | Percentage (%)IQR/SD |
|---|---|---|
| Age (median) | 63 | 51-72 |
| Age (years) | | |
| <30 | 48 | 4.2 |
| 30-39 | 79 | 7.0 |
| 40-49 | 137 | 12.1 |
| 50-59 | 222 | 19.5 |
| 60-69 | 289 | 25.4 |
| 70-79 | 240 | 21.1 |
| ≥80 | 121 | 10.7 |
| Sex | | |
| Female | 551 | 48.5 |
| Male | 585 | 51.5 |
| Residence | | |
| Metropolitan | 253 | 22.3 |
| Non-metropolitan | 883 | 77.7 |
| Work | | |
| Indoor | 876 | 81.9 |
| Outdoor | 193 | 18.1 |
| Missing | 67 | |

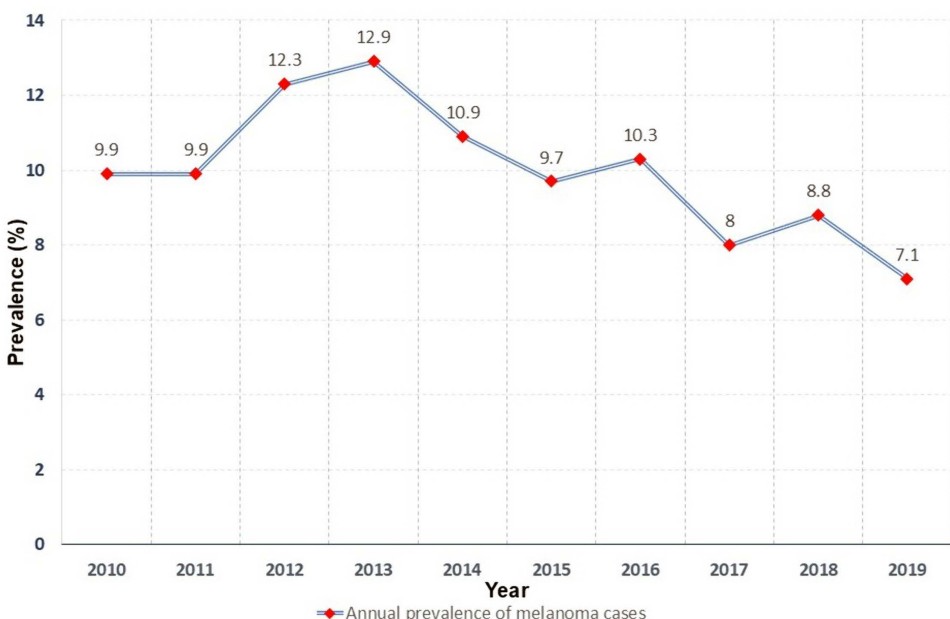

**Fig 2. Annual Prevalence of Melanoma Cases in Our Study, 2010–2019.**

**Table 2. Clinical features of the Peruvian population with melanoma.**

| Variables | Total population (N = 1136) | Percentage (%)/IQR/SD |
|---|---|---|
| Time to diagnosis, months | 12 | 6-24 |
| <6 months | 188 | 16.6 |
| 6–12 months | 208 | 18.4 |
| 12–24 months | 347 | 30.6 |
| 2–3 years | 162 | 14.3 |
| >3 years | 228 | 20.1 |
| Missing | 3 | |
| Location of primary lesion | | |
| Upper extremity | 152 | 14.0 |
| Trunk | 108 | 9.9 |
| Lower extremity | 820 | 75.5 |
| Multiple | 6 | 0.6 |
| Missing | 50 | |
| LDH | | |
| Normal | 387 | 58.4 |
| High | 276 | 41.6 |
| Missing | 473 | |
| Asymmetry of primary lesion | 782 | 68.8 |
| Irregular edges | 690 | 60.9 |
| Heterogenic color | 772 | 68.0 |
| Ulcer | 595 | 52.4 |
| Satellite lesions | 78 | 6.9 |
| Clinical lymph nodes | | |
| Negative | 610 | 53.8 |
| Regional palpable | 206 | 18.2 |
| Matted lymph nodes | 222 | 19.6 |
| Surgery at other institution | 96 | 8.5 |
| Missing | 2 | |

LDH, lactate dehydrogenase.

## Discussion

The present retrospective cohort is one of the largest cohorts in Latin America including 1136 patients diagnosed with melanoma with long-term outcomes in a tertiary referral cancer institute. Our findings indicate participants were mostly middle-aged males who reside in non-metropolitan areas, of whom the majority seek medical care after six months they noticed the lesion usually located on the lower extremities with the presence of ulceration. Most patients within this cohort were AJCC stage III, with a mean Breslow thickness of 8.88 mm, a third of the population required amputation, with the lung as the most common site of metastasis. Greater age, male sex, ulceration, greater mitotic activity, and positive lymph nodes were identified as prognostic factors for poorer EFS and OS. In addition to these factors, mitosis per mm was also associated with worse OS.

In similar cohorts done in the Hispanic population, the mean patient age is around 60–70 years, our data shows that over a quarter of our patients fall into this age range [16–18]. To our knowledge, our cohort contains the largest number of stage III patients in the region, as 60% of participants were AJCC stage III and above. And in general, comparing our work with prior studies from Brazil, Colombia, Argentina, and Ecuador, among others; there is a notable predominance of higher-stage disease in the Peruvian counterpart [18–23].

**Table 3. Pathological and therapeutic features of the Peruvian population with melanoma.**

| Variables | Total population (N = 1136) | Percentage (%)/IQR/SD |
|---|---|---|
| Surgery of primary lesion | | |
| Wide local excision | 482 | 55.4 |
| Other institution, re-excision of margins | 73 | 8.4 |
| Amputation of fingers or toes | 222 | 25.5 |
| Extremity amputation | 26 | 3.0 |
| Biopsy without definitive surgery | 67 | 7.7 |
| Missing | 266 | |
| Lymph node surgery | | |
| None | 173 | 19.4 |
| Sentinel lymph node | 485 | 54.5 |
| Lymph node dissection | 232 | 26.1 |
| Missing | 246 | |
| Total of lymph nodes removed | 5 | 2-13 |
| Total of positive lymph nodes | 1 | 0-2 |
| Surgery for metastasis | 7 | 0.6 |
| Histologic subtype | | |
| Acral lentiginous | 422 | 38.1 |
| Nodular | 154 | 13.9 |
| Other subtypes | 51 | 4.6 |
| Superficial spreading | 36 | 3.2 |
| Lentigo Maligna | 22 | 2.0 |
| Amelanotic | 12 | 1.1 |
| NOS | 412 | 37.2 |
| Missing | 27 | |
| Lymphovascular invasion | | |
| Absent | 776 | 87.9 |
| Present | 107 | f |
| Missing | 253 | |
| Perineural invasion | | |
| Absent | 682 | 85.1 |
| Present | 119 | 14.9 |
| Missing | 335 | |
| Tumor regression | 5 | 0.4 |
| Breslow – mean | 8.88 | 9.59 |
| Mitosis per mm$^2$ – mean | 6.62 | 7.42 |
| T stage | | |
| T1 | 172 | 18.1 |
| T2 | 109 | 11.5 |
| T3 | 205 | 21.6 |
| T4 | 464 | 48.8 |
| Missing | 186 | |
| N stage | | |
| N0 | 334 | 36.9 |
| N1 | 161 | 17.8 |
| N2 | 127 | 14.0 |
| N3 | 282 | 31.2 |

*(Continued)*

**Table 3.** (Continued)

| Variables | Total population (N = 1136) | Percentage (%)/IQR/SD |
|---|---|---|
| Missing | 232 | |
| AJCC Stage | | |
| Stage I | 186 | 16.5 |
| Stage II | 270 | 23.9 |
| Stage III | 412 | 36.5 |
| Stage IV | 260 | 23.0 |
| Missing | 8 | |
| Adjuvant chemotherapy | 152 | 13.4 |
| Adjuvant radiotherapy | 41 | 3.6 |
| Interferon therapy | 155 | 13.6 |
| Site of recurrence of progression | | |
| Lung | 138 | 12.1 |
| Lymph nodes | 106 | 9.3 |
| Skin or subcutaneous tissue | 92 | 8.1 |
| Brain | 82 | 7.2 |
| Liver | 39 | 3.4 |
| Bone | 25 | 2.2 |
| Gastrointestinal | 7 | 0.6 |
| Mediastinum | 6 | 0.5 |
| Suprarenal | 6 | 0.5 |
| Paravesical | 1 | 0.1 |
| Testis | 1 | 0.1 |
| Retroperitoneum | 1 | 0.1 |
| Multiple organs | 21 | 1.8 |

NOS, not otherwise specified; AJCC, American Joint Committee on Cancer

**Table 4. Overall survival and event-free survival of Peruvian patients with melanoma stage I-IV.**

| Survival outcomes | Time periods | | | P value |
|---|---|---|---|---|
| | 3y (%) | 5y (%) | 10y (%) | |
| Event free survival | | | | <0.001 |
| Stage I | 77 | 67 | 30 | |
| Stage II | 51 | 26 | 5 | |
| Stage III | 23 | 12 | 1 | |
| Stage IV | 3 | 1 | 0 | |
| Overall Survival | | | | <0.001 |
| Stage I | 92 | 85 | 70 | |
| Stage II | 73 | 51 | 38 | |
| Stage III | 36 | 22 | 12 | |
| Stage IV | 9 | 6 | 4 | |

When comparing metastatic stages with other developing countries, Bajpai et al. reported on a tertiary cancer center in India shows how 44% of patients presented with metastasis at diagnosis [24]. Smaller Cohorts in Colombian patients also show around 15–30% in Stage IV [19,20,25]. This finding may be explained by the lower rate of dermatological

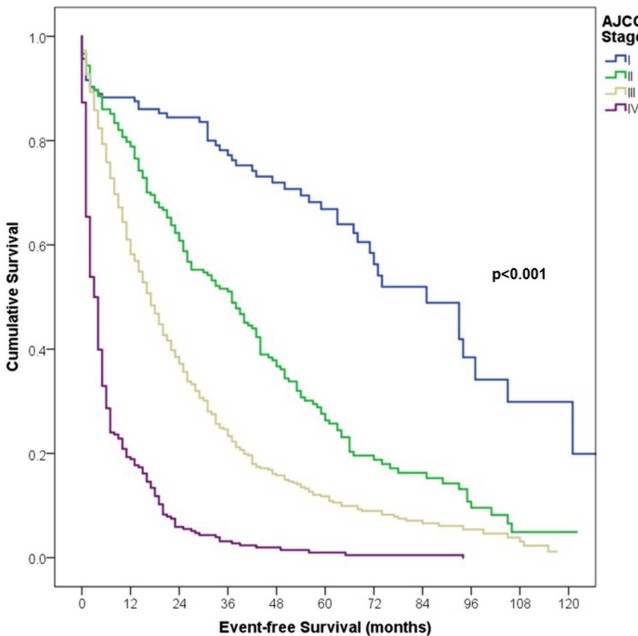

**Fig 3. Comparison of 5-years Event-free Survival according to AJCC stage in patients with melanoma.**

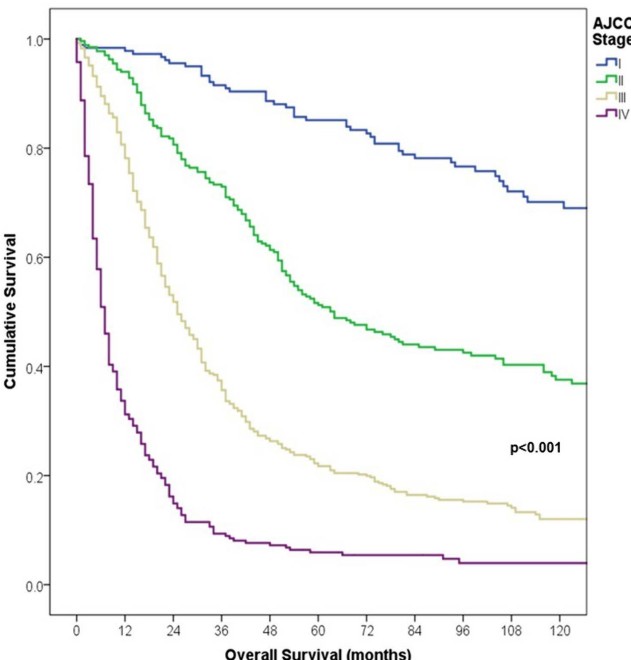

**Fig 4. Comparison of 5-years Overall Survival according to AJCC stage in patients with melanoma.**

**Table 5. Prognostic factors for overall survival and event-free survival of patients with melanoma stage I to IV. <0.05.**

| Variables | Event free survival | | | | | | Overall Survival | | | | | |
| --- | --- | --- | --- | --- | --- | --- | --- | --- | --- | --- | --- | --- |
| | Univariable analysis | | | Multivariable analysis | | | Univariable analysis | | | Multivariable analysis | | |
| | HR | 95%CI | P value | HR | 95%CI | P value | HR | 95%CI | P value | HR | 95%CI | P value |
| Age, years | 1.015 | 1.010-1.019 | <0.001 | 1.020 | 1.013-1.028 | <0.001 | 1.017 | 1.012-1.022 | <0.001 | 1.024 | 1.017-1.031 | <0.001 |
| Sex | | | | | | | | | | | | |
| Female | 1.00 | | | 1.00 | | | 1.00 | | | 1.00 | | |
| Male | 1.522 | 1.329-1.743 | <0.001 | 1.369 | 1.133-1.654 | 0.001 | 1.412 | 1.228-1.624 | <0.001 | 1.201 | 0.990-1.457 | 0.063 |
| Mitosis per mm2 | 1.025 | 1.016-1.034 | <0.001 | 1.012 | 1.001-1.024 | 0.036 | 1.026 | 1.017-1.035 | <0.001 | 1.013 | 1.001-1.025 | 0.032 |
| Breslow | 1.016 | 1.010-1.023 | <0.001 | 1.004 | 0.995-1.012 | 0.396 | 1.019 | 1.012-1.025 | <0.001 | 1.000 | 0.991-1.009 | 0.992 |
| Ulcer | | | | | | | | | | | | |
| No | 1.00 | | | 1.00 | | | 1.00 | | | 1.00 | | |
| Yes | 2.143 | 1.792-2.563 | <0.001 | 1.564 | 1.217-2.009 | <0.001 | 2.496 | 2.073-3.006 | <0.001 | 1.621 | 1.251-2.099 | <0.001 |
| Histologic subtype | | | | | | | | | | | | |
| Acral | 1.00 | | | 1.00 | | | 1.00 | | | 1.00 | | |
| Nodular | 1.265 | 1.026-1.559 | 0.028 | 0.980 | 0.754-1.273 | 0.878 | 1.153 | 0.926-1.438 | 0.204 | 0.960 | 0.734-1.256 | 0.767 |
| NOS and other subtypes | 1.477 | 1.274-1.714 | <0.001 | 1.357 | 1.091-1.687 | 0.006 | 1.357 | 1.164-1.582 | <0.001 | 1.192 | 0.956-1.487 | 0.119 |
| Lymph nodes | | | | | | | | | | | | |
| Negative | 1.000 | | | 1.00 | | | 1.000 | | | 1.00 | | |
| Positive | 2.872 | 2.417-3.413 | <0.001 | 2.580 | 2.091-3.182 | <0.001 | 3.796 | 3.168-4.549 | <0.001 | 3.231 | 2.597-4.020 | <0.001 |
| Site of primary lesion | | | | | | | | | | | | |
| Trunk | 1.000 | | | 1.00 | | | 1.00 | | | 1.00 | | |
| Lower extremity | 1.067 | 0.835-1.364 | 0.602 | 0.732 | 0.507-1.058 | 0.097 | 1.311 | 1.018-1.688 | 0.036 | 0.774 | 0.530-1.131 | 0.186 |
| Upper extremity | 0.699 | 0.513-0.954 | 0.024 | 0.518 | 0.335-0.801 | 0.003 | 0.842 | 0.611-1.161 | 0.295 | 0.607 | 0.386-1.252 | 0.130 |
| Adjuvant chemotherapy | | | | | | | | | | | | |
| No | 1.00 | | | 1.00 | | | 1.000 | | | 1.00 | | |
| Yes | 1.445 | 1.207-1.729 | <0.001 | 1.103 | 0.855-1.421 | 0.451 | 2.009 | 1.665-2.424 | <0.001 | 1.429 | 0.898-1862 | 0.188 |
| Adjuvant radiotherapy | | | | | | | | | | | | |
| No | 1.000 | | | 1.00 | | | 1.000 | | | 1.00 | | |
| Yes | 1.309 | 0.952-1.800 | 0.097 | 1.309 | 0.901-1.903 | 0.158 | 1.738 | 1.253-2.411 | 0.001 | 1.238 | 0.844-1.817 | 0.275 |
| Interferon therapy | | | | | | | | | | | | |
| No | 1.000 | | | 1.00 | | | 1.000 | | | 1.00 | | |
| Yes | 0.753 | 0.623-0.909 | 0.003 | 0.791 | 0.624-1.002 | 0.052 | 0.989 | 0.814-1.201 | 0.909 | 0.940 | 0.739-1.196 | 0.614 |

care-seeking in Hispanic patients be it due to education, economic constraints, or long waiting times for medical appointments due to the overwhelmed health system, factors that delay further diagnosis and management, hence ensuring disease progression [26,27]. In this study, this phenomenon is apparent as over 80% and 70% of patients had a diagnosis over 6 and 12 months, respectively after first noting the lesion and over 20% waited over 3 years for a diagnosis. This is a critical situation that could be classified as worse than other regions of Latin America and should be addressed urgently to improve outcomes [10,28–30].

The decline in melanoma cases seen in our study after 2013 may reflect in referral patterns related to the oncology decentralization in Peru and new regional cancer services such as Plan Esperanza and the opening of regional private cancer centers, rather than a fall in incidence [31]. National and international literature suggest that melanoma incidence and prevalence in Peru is low and has been relatively stable over the last decade [32,33]. Therefore, these findings supports a referral shift explanation more than a real epidemiologic change, acknowledging potential case redistribution across institutions after 2013.

Regarding survival, at three and five years follow-up, overall EFS was 32% and 21%, while the OS was 65% and 53%, respectively. Unsurprisingly, comparison with developed countries showcases the lack of secondary skin preventive policies in Peru, as these same policies when implemented successfully with an effective health care system yield an overall survival of over 90% [1,34,35]. Moreover, drawing comparison from other cancer centers in Latin America [9,19,25], 5-year overall survival remains lower, this finding could be explained by the particularly higher prevalence of the Acral Lentiginous subtype [36], within our patient group compared to other regional cohorts with 38.1% vs. 16–22%, more frequency of ulceration with 52% vs. 31% [25]. As expected, survival was lower due to more aggressive features and a higher risk of disease progression at later stages, plus no availability of targeted therapies and checkpoint inhibitors, especially in advanced stages which are the largest proportion of our population. For instance, advanced cases were treated with a combination of adjuvant chemotherapy, interferon, radiotherapy, and surgical resections for locoregional recurrences. No patients received neoadjuvant therapies and the adjuvant chemotherapy regimen administered was dacarbazine. The impact of these treatments are critical in our outcomes, as reflected in worse OS and EFS rates compared to other countries where immunotherapy available. As our results showed, most patients with advanced disease experienced disease progression and worse survival rates than those diagnosed at early stages.

According to existing literature, the lung was the most common site of metastasis, followed by the brain, liver, and bone similar findings were found in our study [37,38]. We report that 12% of our patients had metastasis to the lung as a relapse followed by the brain with 7% in our population. Lung metastasis may be more prevalent due to its ease of diagnosis, coupled with the increased vascularity and lymphatic return that makes it a frequent metastatic site for most malignant neoplasms [37]. Furthermore, increased tumor-associated macrophages have been identified in the microenvironment of melanoma, and their effect in promoting invasion, extravasation, and colonization of pulmonary tissue [39]. Similarly, brain metastasis in melanoma is not a rare sight as cancer cells are able to cross the blood-brain barrier, and the same mutations in BRAF and NRAS described in lung metastasis, are also present to develop brain metastasis [38,40,41].

Our findings showed that among our population, lymph node involvement was the most important prognostic factor for both OS (HR = 3.32) and EFS (HR = 2.58), followed by ulceration, male sex, greater mitotic index, and greater age. Lymph node metastasis is the cornerstone in the staging of melanoma, which is in unison with Xu et al's study [42], in which lymph node involvement was a key factor in predicting metastasis at the initial stages of the disease [43]. Nowadays the dimension of tumor deposit and extracapsular extension are additional high-risk features, as well as ongoing clinical trials based on the index lymph node to de-escalate complete lymph node dissection after immunotherapy, which unfortunately is still not available in Peru and other developing countries [44–46]. Ulceration is a well-known aggressive feature of melanoma, even incipient ulceration is associated with thicker melanomas, more mitotic activity, more lymphovascular invasion, and satellite lesions than non-ulcerated lesions, therefore worse outcomes [47–49]. Male sex was identified as an independent predictor for worse survival [50,51], Joose et al.[52] propose that behavioral and hormonal differences as well as males have less capacity to neutralize reactive oxygen species may grant females a survival advantage. Likewise, with greater age, tumors tend to be thicker, and have a greater mitotic rate, ulceration and also older patients have been identified to be less likely to notice changes in their skin as well as in discriminating in early changes of pigmented lesions, further delaying treatment [53].

Moreover, prior studies [20,54–56] showed that lesion in the lower extremity was also the most common location of primary lesions in the Hispanic population, however, we found tumors in upper extremities were associated with more favorable EFS. Melanoma in the trunk tends to have a worse prognosis [57] due to them being thicker with a higher mitotic rate whereas lesions in the lower extremities are more prone to experience ulceration which is also related to poor survival. Melanoma in the upper extremities is easier to find for most patients and hence may prompt earlier diagnosis [58]. As for Breslow thickness, a Swedish study [59] on 13,026 patients found that while Breslow thickness is an independent prognostic factor, its significance is reduced when the tumor is ulcerated, and in our study over half of the patients presented in this fashion, the most likely reason why this was not a prognostic factor in our population [60].

Since their advent, ipilimumab, and vemurafenib have become the standard of care for melanoma due to their improvement in survival as well as their promising results in patients with unresectable disease [61]. In spite of being the current standard of care, public institutions in Perú don't have access to them. Moreover, the effects of chemotherapy are minimal and suboptimal compared to newer therapies and in fact [61], adding to their high toxicity rates, outcomes are generally still poor despite the application of multiple cytotoxic agents and combinations [62]. This, coupled with the suboptimal benefit of interferon treatment and ineffective radiotherapy is a challenge to practitioners in treating advanced stages which are 60% of our population. To mitigate this, as explained by Kaufman et al.[63] in their 2016 review, physicians can adopt the combination of carboplatin and paclitaxel or the use of Temozolomide for advanced disease and CNS involvement, respectively. However, this is not the solution to the problem, health care policies are urgently needed from health promotion, and referrals with optimal health system organization, to the acquisition of newer therapies to achieve earlier stages at diagnosis and improve outcomes in later stages [64,65].

Our study presents several strengths such as our definition of OS and EFS, our follow-up started since the treatment date, medical or surgery, which prevents immortal time bias when survival analyses are performed by avoiding observation time without treatment. Therefore, our results demonstrate the long-term effectiveness of melanoma treatment modalities in our population. We included a large number of Peruvian patients characterizing their sociodemographic and clinicopathological profiles. However, our study also has some limitations, absence of treatment options such as immunotherapy or targeted therapies sets up our population to display lower rates of OS. Access to these medications may be available in privately owned or military institutions. In the retrospective design of the study, some medical records were incomplete, or patients were lost to follow-up and excluded from the analysis. We were not able to calculate the compliance to follow-up, which is challenging in retrospective studies due to the reliance in previous data, however, as this study was conducted at a national cancer center, patients were routinely encouraged to adhere to follow-up appointments and were contacted in the event of a missed appointment. We did not include patients diagnosed from 2020 to 2024 in order to ensure at least a five-year follow-up. However, evaluating changes in diagnosis and management during the COVID-19 pandemic is an important topic that would be useful to explore in future research. Molecular profiling by gene expression signatures is not available in public institutions in our setting, therefore diagnosis and classification relied on conventional histopathological evaluation supported when necessary by immunohistochemistry markers, which can limit the classification of melanoma subtypes. Our results should be interpreted with caution if extrapolated to other populations in the setting that this was a single-center experience, despite being a national tertiary referral cancer center. Moreover, we did not include head and neck melanoma patients. Future prospective studies belonging to private centers or centers in rural areas are still needed to fully elucidate the characteristics of melanoma in Peru. These results are useful for Peruvian and Latin American populations but considered with caution if extrapolated to similar populations. New multicentric and international studies are recommended for the generalization of the results.

## Conclusion

Most of the patients were diagnosed at higher stages than similar cohorts within the region, developing, and developing countries, with no access to current standard-of-care medications. Despite similar clinicopathological characteristics to other Latin American studies, EFS and OS rates were much worse than other populations. Prognostic factors from our population are unique with some similarities to other international cohorts. New health care policy changes are urgently needed to optimize diagnosis, treatment, and outcomes of patients with melanoma.

## Supporting information

**S1 File. Deidentified Excel dataset containing clinical and pathologic information on Peruvian patients diagnosed with melanoma.**
(XLSX)

**S2 File.  Comprehensive Data Dictionary for the De-identified Dataset of Peruvian Patients with melanoma.**
(XLSX)

## Acknowledgments

The authors thank the Universidad Científica del Sur for their support in the publication of this research/project.

## Author contributions

**Conceptualization:** Gonzalo Ziegler Rodriguez, Gabriel de la Cruz-Ku, Anshumi Desai, Luis Piedra Delgado, Silvana Maldonado, Jiddu Antonio Guart, Camila Franco, Sheyla Diaz-Mora, Sheila Vilchez Santillan, Jorge Dunstan Yataco, Juan Haro-Varas, Jose Galarreta Zegarra, Sandro Casavilca Zambrano, Jose Cotrina Concha.

**Data curation:** Gonzalo Ziegler Rodriguez, Gabriel de la Cruz-Ku, Silvana Maldonado, Sheyla Diaz-Mora, Sheila Vilchez Santillan, Juan Haro-Varas, Jose Galarreta Zegarra, Sandro Casavilca Zambrano, Jose Cotrina Concha.

**Formal analysis:** Gonzalo Ziegler Rodriguez, Gabriel de la Cruz-Ku, Luis Piedra Delgado, Sheila Vilchez Santillan.

**Funding acquisition:** Gonzalo Ziegler Rodriguez, Gabriel de la Cruz-Ku, Sheila Vilchez Santillan.

**Investigation:** Gonzalo Ziegler Rodriguez, Gabriel de la Cruz-Ku, Anshumi Desai, Luis Piedra Delgado, Silvana Maldonado, Jiddu Antonio Guart, Camila Franco, Sheyla Diaz-Mora, Sheila Vilchez Santillan, Juan Haro-Varas, Jose Galarreta Zegarra, Sandro Casavilca Zambrano.

**Methodology:** Gonzalo Ziegler Rodriguez, Gabriel de la Cruz-Ku, Luis Piedra Delgado, Silvana Maldonado, Jiddu Antonio Guart, Camila Franco, Jorge Dunstan Yataco, Jose Galarreta Zegarra, Sandro Casavilca Zambrano.

**Project administration:** Gonzalo Ziegler Rodriguez, Gabriel de la Cruz-Ku, Camila Franco, Jose Galarreta Zegarra, Sandro Casavilca Zambrano, Jose Cotrina Concha.

**Resources:** Gonzalo Ziegler Rodriguez, Gabriel de la Cruz-Ku, Jorge Dunstan Yataco.

**Software:** Gonzalo Ziegler Rodriguez, Gabriel de la Cruz-Ku, Anshumi Desai, Sheyla Diaz-Mora.

**Supervision:** Gonzalo Ziegler Rodriguez, Gabriel de la Cruz-Ku, Anshumi Desai, Jiddu Antonio Guart, Sheyla Diaz-Mora, Sheila Vilchez Santillan, Jorge Dunstan Yataco, Juan Haro-Varas, Jose Galarreta Zegarra, Sandro Casavilca Zambrano, Jose Cotrina Concha.

**Validation:** Gonzalo Ziegler Rodriguez, Gabriel de la Cruz-Ku, Anshumi Desai, Camila Franco, Sheyla Diaz-Mora, Sheila Vilchez Santillan, Jose Galarreta Zegarra, Jose Cotrina Concha.

**Visualization:** Gonzalo Ziegler Rodriguez, Gabriel de la Cruz-Ku, Anshumi Desai, Luis Piedra Delgado, Silvana Maldonado, Jiddu Antonio Guart, Camila Franco, Sheyla Diaz-Mora, Sheila Vilchez Santillan, Jorge Dunstan Yataco, Juan Haro-Varas, Jose Galarreta Zegarra, Sandro Casavilca Zambrano, Jose Cotrina Concha.

**Writing – original draft:** Gonzalo Ziegler Rodriguez, Gabriel de la Cruz-Ku, Anshumi Desai, Luis Piedra Delgado, Silvana Maldonado, Jiddu Antonio Guart, Camila Franco, Sheyla Diaz-Mora, Sheila Vilchez Santillan, Jorge Dunstan Yataco, Juan Haro-Varas, Jose Galarreta Zegarra, Sandro Casavilca Zambrano, Jose Cotrina Concha.

**Writing – review & editing:** Gonzalo Ziegler Rodriguez, Gabriel de la Cruz-Ku, Anshumi Desai, Luis Piedra Delgado, Silvana Maldonado, Jiddu Antonio Guart, Camila Franco, Sheyla Diaz-Mora, Sheila Vilchez Santillan, Jorge Dunstan Yataco, Juan Haro-Varas, Jose Galarreta Zegarra, Sandro Casavilca Zambrano, Jose Cotrina Concha.

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
