## [Editor Report · Decision Letter 0]

21 Mar 2025

PONE-D-25-11273Melanoma in Peru: 1000 patients and 10 years of experiencePLOS ONE

Dear Dr. de la Cruz-Ku,

Thank you for submitting your manuscript to PLOS ONE. After careful consideration, we feel that it has merit but does not fully meet PLOS ONE’s publication criteria as it currently stands. Therefore, we invite you to submit a revised version of the manuscript that addresses the points raised during the review process.

We look forward to receiving your revised manuscript.

Kind regards,

Suvendu Maji, MBBS.MS(GENERAL SURGERY).DNB(SURGICAL ONCOLOGY)

Academic Editor

PLOS ONE

2. We note that there is identifying data in the Supporting Information file < Melanoma DATA FINAL 10-16-24.xlsx>. Due to the inclusion of these potentially identifying data, we have removed this file from your file inventory. Prior to sharing human research participant data, authors should consult with an ethics committee to ensure data are shared in accordance with participant consent and all applicable local laws.

-Location data

Please remove or anonymize all personal information, ensure that the data shared are in accordance with participant consent, and re-upload a fully anonymized data set. Please note that spreadsheet columns with personal information must be removed and not hidden as all hidden columns will appear in the published file.

Additional Editor Comments:

Excellent work and I congratulate the authors for their hardwork.

Please clarify the following..

1.How follow up was done for 10 long years?

2.what was the patient compliance to follow up and attrition rate?

3.How the patients were managed( advanced cases) as many of them would have progressed in absence of immunotherapy.

4.Since AJCC ( 8th) edition has been used..how have the older patients (5 years back cases) classified??Have this change been taken care of while the data has been compiled?

---

## [Author Response · Author response to Decision Letter 1]

8 Apr 2025

Reviewer’s comment:

Excellent work and I congratulate the authors for their hardwork.

Authors’ response:

Thank you very much for the kind words of the reviewer, we really appreciate it.

Reviewer’s comment:

Please clarify the following..

1.How follow up was done for 10 long years?

Authors’ response:

Thank you for this comment, our follow up was done according to NCCN guidelines. After completion of treatment, for stages I-IIA, patients were followed up every 6 months for five years and then annually as clinically indicated. For patients with stage IIB to IV, these were followed every three months for a total of two years, then every six months for three additional years, then annually depending on the clinical status of the patient. For patients with stage IIB to IV, every patient has scheduled computed tomography (CT) scans with intravenous contrast and ultrasounds of the adjacent lymph node regions every 12 months for three years, then additional CT scans or ultrasounds are performed if the patient presents symptoms. We have included these statements in our materials and methods.

Reviewer’s comment:

2.what was the patient compliance to follow up and attrition rate?

Authors’ response:

Thank you very much for this comment to improve our manuscript. The attrition rate was 5.1% at 5 years, and 17.5% at 10 years. The compliance to follow-up is challenging to measure in a retrospective study such as number of visits that they attended. Despite we do not have this information in our study, the fact that this was done at a national cancer center, patients are encouraged to adhere to follow-up appointments and were contacted in the event of a missed appointment. These statements have been included in our materials and methods and limitations.

Reviewer’s comment:

3.How the patients were managed( advanced cases) as many of them would have progressed in absence of immunotherapy.

Authors’ response:

Thank you for this comment. Due to unavailability of immunotherapy in our country, advanced cases were treated with a combination of chemotherapy, interferon, and radiotherapy, and surgical resections for locoregional recurrences. No patients received neoadjuvant therapies and the adjuvant chemotherapy regimen administered was dacarbazine. Therefore, we had worse survival outcomes (overall and event free survival) compared to other countries with immunotherapy available. Indeed, as our results showed, most patients with advanced disease experienced disease progression and worse survival rates than those diagnosed at early stages. We have included these statements in our discussion.

Reviewer’s comment:

4.Since AJCC ( 8th) edition has been used..how have the older patients (5 years back cases) classified??Have this change been taken care of while the data has been compiled?

Authors’ response:

We appreciate the comment of the reviewer. We agree that we collected data from patients that were diagnosed before the AJCC 8th edition was published. For this reason when medical records were reviewed, all the cases were re-classified according to the AJCC 8th edition. Each case was reviewed with clinical and pathological data to perform an accurate staging classification. We have included this in our materials and methods.

---

## [Decision Letter · Decision Letter 1]

6 Aug 2025

PONE-D-25-11273R1Melanoma in Peru: 1000 patients and 10 years of experience

PLOS ONE

Dear Dr. de la Cruz-Ku,

Thank you for submitting your manuscript to PLOS ONE. After careful consideration, we feel that it has merit but does not fully meet PLOS ONE’s publication criteria as it currently stands. Therefore, we invite you to submit a revised version of the manuscript that addresses the points raised during the review process.

Please respond to all reviewer comments and revise the manuscript accordingly. On my end, I kindly ask you to review the submitted table (Melanoma Data Final); some cells display the value '#NULL!'. Please clarify whether these represent missing data or if they are due to an error. Additionally, include a comprehensive data dictionary for all variables used.

I also understand the importance of having at least a five-year follow-up period (2019–2024), which may justify the end of the inclusion period; however, this should be clearly stated and discussed in the manuscript. Furthermore, the survival curves must include the associated p-values to allow assessment of whether there are statistically significant differences between groups.

We look forward to receiving your revised manuscript.

Kind regards,

Alexis G. Murillo Carrasco

Academic Editor

PLOS ONE

Journal Requirements:

Reviewers' comments:

Reviewer's Responses to Questions

**Comments to the Author**

1. If the authors have adequately addressed your comments raised in a previous round of review and you feel that this manuscript is now acceptable for publication, you may indicate that here to bypass the “Comments to the Author” section, enter your conflict of interest statement in the “Confidential to Editor” section, and submit your "Accept" recommendation.

Reviewer #1: (No Response)

Reviewer #2: All comments have been addressed

Reviewer #3: All comments have been addressed

Reviewer #4: All comments have been addressed

2. Is the manuscript technically sound, and do the data support the conclusions?

Reviewer #1: Yes

Reviewer #2: Yes

Reviewer #3: Yes

Reviewer #4: Yes

3. Has the statistical analysis been performed appropriately and rigorously? 

Reviewer #1: No

Reviewer #2: Yes

Reviewer #3: Yes

Reviewer #4: Yes

4. Have the authors made all data underlying the findings in their manuscript fully available?

Reviewer #1: No

Reviewer #2: Yes

Reviewer #3: Yes

Reviewer #4: Yes

5. Is the manuscript presented in an intelligible fashion and written in standard English?

Reviewer #1: Yes

Reviewer #2: Yes

Reviewer #3: Yes

Reviewer #4: Yes

6. Review Comments to the Author

Reviewer #1: Why was the data only collected up to 2019? It would be reasonable to expect inclusion of more recent data. It may be interesting to observe any changes that may have occurred following the COVID-19 pandemic.

Could you show the prevalence over the years?

Also, the terms univariate/multivariate analyses would be more accurately described as univariable/multivariable analyses in this context.

Although Kaplan-Meier analysis was mentioned, the results were not presented.

Reviewer #2: The authors present a very well organized and analyzed retrospective cohort study of melanoma in Peru, all of the issues raised by previous reviewers were adequately addressed.

Reviewer #3: (No Response)

Reviewer #4: This is a well done study that sheds light on the sociodemographic, clinicopathological features, and long-term outcomes such as event-free survival (EFS) and overall survival (OS) Peruvian population with melanoma.

However, there were some limitations of this study that should be discussed by the authors in the interpretation of their findings. The authors should indicate in the Materials and methods section how the specimens was confirmed by the Pathology department of the institute (Breslow thickness exclusively and/or IHC markers ?) for a more precise assessment of the tumoral thickness. The use of immunohistochemical markers for melanoma diagnosis rather than gene expression signatures data for classification of melanoma subtypes was an obvious limitation of this study that should be considered in the Discussion section.

7. PLOS authors have the option to publish the peer review history of their article (what does this mean? ). If published, this will include your full peer review and any attached files.

**Do you want your identity to be public for this peer review?** For information about this choice, including consent withdrawal, please see our Privacy Policy .

Reviewer #1: No

Reviewer #2: **Yes: ** Benjamin Benzon

Reviewer #3: No

Reviewer #4: No

---

## [Author Response · Author response to Decision Letter 2]

25 Aug 2025

EDITOR

Reviewer’s comment:

Please respond to all reviewer comments and revise the manuscript accordingly. On my end, I kindly ask you to review the submitted table (Melanoma Data Final); some cells display the value '#NULL!'. Please clarify whether these represent missing data or if they are due to an error. Additionally, include a comprehensive data dictionary for all variables used.

Authors’ response:

We appreciate the comments provided. #NULL! Means that these values were missing data, we have kept them as NULL rather than leaving those values in blank. We have also mentioned in the methods section that we have excluded these missing values from the analysis. We have also included a comprehensive data dictionary for all variables used.

Reviewer’s comment:

I also understand the importance of having at least a five-year follow-up period (2019–2024), which may justify the end of the inclusion period; however, this should be clearly stated and discussed in the manuscript.

Authors’ response:

We appreciate this comment to improve our manuscript. We have added this statement to our materials and methods.

Reviewer’s comment:

Furthermore, the survival curves must include the associated p-values to allow assessment of whether there are statistically significant differences between groups.

Authors’ response:

We appreciate this valuable comment and have added the p-values to the survival curve figures.

REVIEWER 1

Reviewer’s comment:

Reviewer #1: Why was the data only collected up to 2019? It would be reasonable to expect inclusion of more recent data. It may be interesting to observe any changes that may have occurred following the COVID-19 pandemic.

Authors’ response:

We appreciate the comment of the reviewer, the main reason was to have at least five-year follow up period, being the last year included 2019. We agree with the reviewer’s observation about the changes during COVID-19 pandemic, we have included this in our limitation and proposed as a future study.

Reviewer’s comment:

Could you show the prevalence over the years?

Authors’ response:

We thank the reviewer for the suggestion. We have now included a figure (Figure 2) showing the yearly prevalence of cases from 2010 to 2019, which allows easier comparison of prevalence across the years. We have also included these findings in our results section, a figure (Figure 2) and have added a paragraph to discuss these findings.

Reviewer’s comment:

Also, the terms univariate/multivariate analyses would be more accurately described as univariable/multivariable analyses in this context.

Authors’ response:

We thank the reviewer for this valuable comment, we have corrected this statement throughout our manuscript.

Reviewer’s comment:

Although Kaplan-Meier analysis was mentioned, the results were not presented.

Authors’ response:

We appreciate the reviewer’s comment. These documents were originally uploaded as images; we have now included the images within the manuscript, separate from the uploaded documents, to improve understanding. These results were mentioned in page 12 before Table 4.

REVIEWER 2

Reviewer’s comment:

Reviewer #2: The authors present a very well organized and analyzed retrospective cohort study of melanoma in Peru, all of the issues raised by previous reviewers were adequately addressed.

Authors’ response:

We appreciate the comments of the reviewer.

REVIEWER 3

REVIEWER 3

Reviewer #3: (No Response)

REVIEWER 4

Reviewer’s comment:

Reviewer #4: This is a well done study that sheds light on the sociodemographic, clinicopathological features, and long-term outcomes such as event-free survival (EFS) and overall survival (OS) Peruvian population with melanoma.

However, there were some limitations of this study that should be discussed by the authors in the interpretation of their findings. The authors should indicate in the Materials and methods section how the specimens was confirmed by the Pathology department of the institute (Breslow thickness exclusively and/or IHC markers ?) for a more precise assessment of the tumoral thickness. The use of immunohistochemical markers for melanoma diagnosis rather than gene expression signatures data for classification of melanoma subtypes was an obvious limitation of this study that should be considered in the Discussion section.

Authors’ response:

We appreciate the reviewer’s valuable comments to improve our manuscript. Specimens were confirmed by the Pathology Department of our institute, with all cases reviewed by at least two pathologists. Tumor thickness was assessed according to Breslow depth. In cases where histopathological evaluation alone was insufficient, immunohistochemical (IHC) markers were employed to support the diagnosis of melanoma. Unfortunately, gene expression signatures are not available in public institutions in our country. These statements have been added to the Materials and Methods section, and the lack of molecular classification has been acknowledged as a limitation in the Discussion section.

---

## [Decision Letter · Decision Letter 2]

1 Sep 2025

Melanoma in Peru: 1000 patients and 10 years of experience

PONE-D-25-11273R2

Dear Dr. de la Cruz-Ku,

We’re pleased to inform you that your manuscript has been judged scientifically suitable for publication and will be formally accepted for publication once it meets all outstanding technical requirements.

Kind regards,

Alexis G. Murillo Carrasco

Academic Editor

PLOS ONE

Additional Editor Comments (optional):

Reviewer #1:

Reviewer #3:

Reviewers' comments:

Reviewer's Responses to Questions

**Comments to the Author**

1. If the authors have adequately addressed your comments raised in a previous round of review and you feel that this manuscript is now acceptable for publication, you may indicate that here to bypass the “Comments to the Author” section, enter your conflict of interest statement in the “Confidential to Editor” section, and submit your "Accept" recommendation.

Reviewer #1: All comments have been addressed

Reviewer #3: All comments have been addressed

2. Is the manuscript technically sound, and do the data support the conclusions?

Reviewer #1: (No Response)

Reviewer #3: Yes

3. Has the statistical analysis been performed appropriately and rigorously? 

Reviewer #1: (No Response)

Reviewer #3: Yes

4. Have the authors made all data underlying the findings in their manuscript fully available?

Reviewer #1: (No Response)

Reviewer #3: No

5. Is the manuscript presented in an intelligible fashion and written in standard English?

Reviewer #1: (No Response)

Reviewer #3: Yes

6. Review Comments to the Author

Reviewer #1: All my concerns are addressed.

Reviewer #3: The manuscript is well written, scientifically sound, and provides valuable insights. The methodology is appropriate, and the conclusions are supported by the data presented. I did not identify any concerns regarding dual publication, research ethics, or publication ethics.

7. PLOS authors have the option to publish the peer review history of their article (what does this mean? ). If published, this will include your full peer review and any attached files.

**Do you want your identity to be public for this peer review?** For information about this choice, including consent withdrawal, please see our Privacy Policy .

Reviewer #1: No

Reviewer #3: No

---

## [Editor Report · Acceptance letter]

PONE-D-25-11273R2

PLOS ONE

Dear Dr. de la Cruz-Ku,

I'm pleased to inform you that your manuscript has been deemed suitable for publication in PLOS ONE. Congratulations! Your manuscript is now being handed over to our production team.

Kind regards,

on behalf of

Dr. Alexis G. Murillo Carrasco

Academic Editor

PLOS ONE